# Sugar-sweetened beverage intake and convenience store shopping as mediators of the food insecurity–Tooth decay relationship among low-income children in Washington state

**Courtney M. Hill**[1,2]*, **Donald L. Chi**[1,3], **Lloyd A. Mancl**[1], **Jessica C. Jones-Smith**[2,3], **Nadine Chan**[2,4], **Brian E. Saelens**[5,6], **Christy M. McKinney**[1,5,6]

1 Department of Oral Health Sciences, University of Washington, Seattle, WA, United States of America,
2 Department of Epidemiology, University of Washington, Seattle, WA, United States of America,
3 Department of Health Systems and Population Health, University of Washington, Seattle, WA, United States of America, 4 Public Health-Seattle & King County, Assessment, Policy, Development and Evaluation Division, Seattle, WA, United States of America, 5 Seattle Children's Research Institute, Seattle, WA, United States of America, 6 Department of Pediatrics, University of Washington, Seattle, WA, United States of America

* chill7@uw.edu

## Abstract

### Introduction

There are oral health disparities in the U.S. and children in food-insecure households have a higher burden of tooth decay. Identifying the mechanisms underlying the food insecurity–tooth decay relationship could inform public health interventions. This study examined how sugar-sweetened beverage (SSB) intake and frequent convenience store shopping mediated the food insecurity–tooth decay relationship for lower-income children.

### Materials and methods

Cross-sectional study data included a household survey, beverage questionnaire, and dental examination. The sample included 452 lower-income, racially-diverse, child-caregiver dyads in 2018 from King County in Washington state. The exposure was household food insecurity, the outcome was untreated decayed tooth surfaces, and the proposed mediators were SSB intake and frequent convenience store shopping (≥2 times/week). Causal mediation analyses via the potential outcomes framework was used to estimate natural indirect and direct effects.

### Results

Fifty-five percent of participants were in food-insecure households, the mean number of decayed tooth surfaces among children was 0.87 (standard deviation [SD] = 1.99), the mean SSB intake was 17 fluid ounces (fl/oz)/day (SD = 35), and 18% of households frequently shopped at a convenience store. After adjusting for confounders, household food

**Data Availability Statement:** All relevant data are within the paper and its Supporting Information files.

**Funding:** This work was supported by funds from Seattle Children's Research Institute and ARCORA – The Foundation of Delta Dental of Washington (CMM)(https://arcorafoundation.org/). Funding for the Sugary Beverage Tax evaluation in Seattle was provided through ordinance by the City of Seattle (Seattle.gov)(JJS;NC;BES). Elected representatives and city staff did not influence the evaluation findings or interpretation of findings. The funders had no role in study design, data collection and analysis, decision to publish, or preparation of the manuscript.

**Competing interests:** The authors have declared that no competing interests exist.

insecurity and log-transformed SSB intake (fluid ounces/day) were positively associated with decayed tooth surfaces, but not at the a α = 0.05 level (mean ratio [MR] 1.60; 95% confidence interval [CI] 0.89, 2.88; p = .12 and MR 1.16; 95% CI 0.93, 1.46; p = .19, respectively). Frequent convenience store shopping was associated with 2.75 times more decayed tooth surfaces (95% CI 1.61, 4.67; p < .001). SSB intake mediated 10% of the food insecurity–tooth decay relationship (p = .35) and frequent convenience store shopping mediated 22% (p = .33).

## Conclusions

Interventions aimed at addressing oral health disparities in children in food-insecure households could potentially focus on reducing intake of SSBs and improving access to healthful foods in lower-income communities.

## Introduction

Tooth decay is highly prevalent among U.S. children and disproportionately impacts socioeconomically-disadvantaged children [1]. Hispanic children, Black children, and children in low-income households are at particularly high-risk for tooth decay [2,3]. The high burden of tooth decay in these subgroups stem from underlying structural and systemic inequalities such as poor access to dental insurance, quality dental care, and nutritious foods [4], inequalities in wealth and income [5], and racism [6]. Consequences of childhood tooth decay include poor overall health, pain, missed school days, lower quality of life, and costly health care needs [7–9]. Previous studies have reported that household food insecurity is an important risk factor for tooth decay in children [10–15]. According to the United States Department of Agriculture, food is a household-level economic and social condition of limited or uncertain access to adequate food [16]. Understanding the mechanisms underlying the household food insecurity–tooth decay relationship could identify the most relevant pathways to target when developing public health interventions aimed at preventing tooth decay and improving the oral health and overall health of low-income children.

Few studies have considered the pathway for the household food insecurity–tooth decay relationship. The one study aimed to examine diet quality as a potential mediator using a large nationally-representative sample of U.S. children but did not conduct mediation analysis because the association between overall diet quality and tooth decay was not statistically significant, in accordance with the traditional Baron and Kenny approach to mediation [14]. Other potential mediators include specific components of diet (e.g., sugar-sweetened beverage [SSB] intake) and behaviors associated with diet quality (e.g., food store shopping), although these have not been considered before in the literature. Excess intake of added sugars, including those from SSBs, is causally associated with the development of tooth decay, because sugar acts as a substrate for oral bacteria that cause tooth decay [17,18]. Children with low socioeconomic status (SES) and children in food-insecure households tend to have higher levels of SSB intake compared to children with higher SES and children in food-secure households [17,19–23].

Frequent convenience store shopping is a potential mediator of the relationship between food insecurity and tooth decay because convenience stores are an important setting where excess added sugars and SSBs are purchased. Research has shown that the most common convenience store purchase is soda [24,25], convenience store purchases frequently exceed

recommendations for added sugar intake [25], and convenience stores are the most common setting that low SES populations purchase SSBs [26,27]. In addition, food-insecure households are more likely to shop at convenience stores [28–30]. This results from both limited access to other food store options and from food assistance program cycling where shopping behaviors shift to affordable food options when food money runs out [31]. If analyses show that convenience store shopping is a relevant mediator, public health interventions could modify the food environment by improving access to affordable and healthy foods in convenience stores and by changing shopping and purchasing behaviors in convenience stores.

In this study, we leverage causal mediation methods to better understand the mechanisms underlying the household food insecurity–tooth decay relationship. The aim was to determine whether SSB intake and frequent convenience store shopping mediate the food insecurity–tooth decay relationship for lower-income children. Mediation analysis was used in this analysis because it is a tool used to discover and test possible causal relationships by teasing apart direct effects of an exposure on an outcome from indirect effects that occur through a third mediator variable. Findings can inform public health efforts aimed at eliminating oral health disparities among children with low SES.

## Materials and methods

### Study design and data source

This study used cross-sectional data from a racially-diverse sample of children and caregivers from lower-income households in Seattle and South King County in Washington state. The sample was based on the SeaSAW study (a cohort study designed to evaluate the Seattle SSB tax) [32]. Data collection for SeaSAW including the household survey and beverage consumption questionnaire began in October 2017 and was completed in January 2018 with child dental examinations occurring between April and June 2018. Data were collected in a period for which the enactment of the SSB tax would not influence the variables of interest (the tax went into effect January 2018). Written informed consent was obtained from the caregivers of study participants. The study was approved by Seattle Children's Hospital Institutional Review Board. Authors had access to information that identify individual participants during and after data collection.

### Study population

Children-parent dyads who participated in the SeaSAW study who met study eligibility criteria for the dental examination were invited to participate in this additional oral health study. We also recruited friends and acquaintances of SeaSAW participants, and other members of Spanish and Somali communities with the help of community organizers. There were five eligibility criteria for participation in the additional study: (1) the child was in 1st to 9th grade (5 to 16 years old); (2) the child consumed SSBs based on parent report; (3) the annual household income was <312% of the Federal Poverty Level (FPL); (4) the caregiver was ≥18 years old; and (5) the caregiver spoke English, Spanish, Somali or Vietnamese. The screening item about SSBs was, "Does your child ever drink sugary beverages like: regular soda/pop (such as Coke or Sprite), fruit-flavored drinks (like Sunny Delight), coffee or tea drinks with added sugar (like Starbucks Frappucinnos, Arizona Iced Tea, Chai Tea, bubble tea), or regular sports drinks or energy drinks (such as Gatorade or Red Bull)?". This screening item was different than the main study measure of SSB intake which only captures intake in the past 30 days so it possible that some participants had an affirmative response to the screening item but did not report SSB in the past 30 days in later study measured. Defining low-income as <312% of the Federal Poverty Level corresponds to cut-off points used by Washington state public health insurance

programs. The SeaSAW study had slightly different child age criterion; namely, children were required to be either 7–10 or 12–17 years old, but otherwise had the same inclusion criteria [33]. We recruited and limited the sample to be from communities with fluoridated water to eliminate the possibility that community water fluoridation affected study results because water fluoridation is known to weaken the positive associations between added sugar intake and tooth decay [34–36].

## Study variables

**Exposure.** The study exposure, food insecurity, was assessed with a 2-item screener using a "last 12-months" reference period [37]. Compared to the U.S. Household Food Security Survey Module: Six-Item Short Form [38], the 2-item screener has 95.4% sensitivity and 83.5% specificity [39]. Households with an affirmative response to either item were considered food-insecure.

**Mediators.** Potential mediators of interest were child SSB intake and frequent convenience store shopping (**Fig 1**). SSB intake was measured with a validated beverage intake questionnaire (BEV-Q) that was modified to differentiate between sugary beverages that were soon to be taxed within Seattle (e.g., bottled sports drinks with sugar versus sports drink powder with sugar used to constitute the beverage) [40–42]. The BEV-Q asks about intake (typical frequency [ranging from never or less than 1 time/week to 3+ times/day] and volume [ranging from less than 6 fluid ounces to more than 20 fluid ounces] in the past 30 days) of different types of sugary and non-sugary beverages. The caregiver completed the BEV-Q regarding their child's beverage consumption for children <11 years old. Children ≥ 11 years old completed their own BEV-Q regarding their consumption. The BEV-Q assessment of SSB intake was tested in adolescents and shown to have high validity against multiple 24-hour dietary recalls, high reliability across repeat measurements, and readability scores appropriate for individuals with a fourth grade education or higher [40–42]. To score the BEV-Q, frequency of intake is converted to the unit of times/day, then multiplied by the typical amount consumed

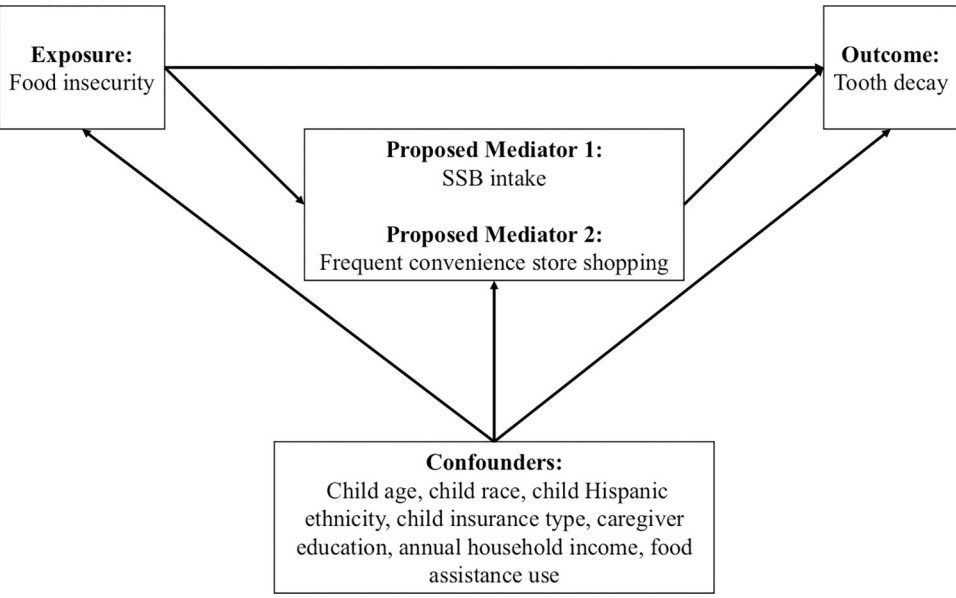

**Fig 1. Conceptual model for potential mediators in the food insecurity–tooth decay relationship for children, aged 5 to 16 years, in Seattle and South King County, Washington 2018.**

to provide average daily beverage intake in fluid ounces (fl oz) [41]. Intake of each SSB item (including SSBs that were and were not subject to the Seattle SSB tax) was summed to create one variable for total SSB intake in units of fl oz/day. SSB intake was then assessed two ways: log-transformed SSB intake (natural log(x+1)) to account for a right skew and to include zero values, and as a categorical variable (0 fl oz/day, >0 fl oz/day to 8 fl oz/day, >8 fl oz/day to 16 fl oz/day, and >16 fl oz/day). SSB intake was also examined as a binary variable in sensitivity analyses (>0 fl oz/day vs. 0 fl oz/day).

Frequency of convenience store shopping was measured with five possible responses (never, <1 time/month, 1 time/month, 1 time every other week, 1 time/week, ≥2 times/week). It was categorized into three groups (never or <1 time/month, 1 time/month to 1 time/week, and ≥2 times/week. Frequent convenience store shopping was defined as ≥2 times/week.

**Outcome.** The study outcome was tooth decay operationalized as a count variable based on the number of surfaces with untreated decay on primary or permanent teeth. Untreated tooth decay was the main outcome because it reflects current disease status; including teeth missing due to disease and filled surfaces may capture disease that occurred before the exposures and mediators of interest in this study. However, as a sensitivity analyses, the number of decayed, missing, and filled tooth surfaces (DMFS) was also assessed as an outcome. The International Caries Detection and Assessment System (ICDAS, version 2 or 'ICDAS') was followed for dental examinations [43,44]. A lesion code of ≥3 was the cut off to identify decayed tooth surfaces.

All dental examiners (dentist or dental hygienist) were calibrated in the ICDAS during an intensive 3-day training class which included didactic materials and practice examinations. Duplicate examinations on a sample of study participants were conducted to estimate inter-rater reliability. About 55% of the sample received an examination by both a gold standard examiner and another examiner. Inter-rater agreement was high, >98% surfaces were correctly coded. Cohen's kappa for all carious surfaces was 0.85 (0.87 for primary and 0.73 for permanent surfaces). These values are classified as substantial to almost perfect [45] and acceptable and like other studies using ICDAS [44,46].

**Confounders.** Confounders were variables hypothesized to be associated with the exposure and outcome that do not lie on the causal pathway between food insecurity and tooth decay [47]. We also identified exposure-mediator confounders and mediator-outcome confounders. The minimum set of confounders to adjust for were identified using directed acyclic graphs (DAGs) (**S1 File**). We adjusted models for the following potential confounders: child age, child race, child ethnicity, child insurance type, caregiver highest level of education, annual household income, and food assistance use (defined as participation in either Supplemental Nutrition Assistance Program [SNAP] or Special Supplemental Nutrition Program for Women, Infants, and Children [WIC].). These variables were based on the caregiver report on the household survey. Although race and ethnicity were used as confounders in this study, we acknowledge that it is racism and discrimination, which were not measured, and not race and ethnicity, that shape an individual's exposures and opportunities [48,49]. In this study, exposure to racism and discrimination are likely determinants of food environment, income, food insecurity, education, and employment opportunities [48,49].

## Statistical analyses

We generated descriptive statistics (mean, standard deviation [SD] or n, %) for participant characteristics. Because the distributions of both tooth decay and SSB intake were skewed, we also report median and interquartile range (IQR). In separate Poisson regression models, we examined the associations between 1) household food insecurity and number of decayed tooth

surfaces, 2) SSB intake and number of decayed tooth surfaces, and 3) convenience store shopping number of decayed tooth surfaces. Each model was adjusted for confounders and confidence intervals (CI) were generated with robust standard errors. Because of the age range of the study population, there was a combination of primary, permanent, and mixed dentitions, so all multivariable analyses modeling tooth decay as an outcome included adjustment for the number of tooth surfaces as a covariate in the regression model.

We used causal mediation analysis based on the potential outcomes framework [50,51] to decompose indirect and direct pathways between household food insecurity and tooth decay. We estimated two mediation effects: 1) the natural indirect effect which measures the effect of the exposure (e.g., food security) that operates through the mediator (e.g., SSB intake or frequent convenience store shopping); and 2) the natural direct effect, which measures the impact of the exposure if it did not cause the mediator [52]. The proportion mediated, which can be interpreted as the overall impact of food insecurity on tooth decay attributable to the impact of food insecurity on the mediator, was estimated and expressed as a percentage.

All mediation estimates were generated using the quasi-Bayesian approach developed by Imai et al. that uses Monte Carlo simulations based on normal approximations [50,51]. In the quasi-Bayesian approach, the natural indirect effect and natural direct effect are calculated by estimating the difference in the observed and counterfactual outcomes across many simulations. We used 1,000 simulations to generate the estimates and robust standard errors were used to generate 95% CI. The mediation estimates for are reported on the additive scale (mean differences in the number of decayed tooth surfaces) at the mean level for quantitative confounders or the most frequent level for categorical confounders. For the mediation analysis, we fit a linear regression model for natural log-transformed SSB intake and a Poisson regression model for frequent convenience store shopping conditional on household food insecurity and confounders. We then fit a Poisson regression model for number of untreated decayed surfaces conditional on household food insecurity, the mediator of interest, and confounders (S2 File).

There was minor missingness in the analytical variables (<6%); observations with missing data for the analytical variables were removed for a complete-case analysis. All analysis was conducted in R version 4.03 using the mediation package [50,53].

**Statistical assumptions and sensitivity analyses.** Causal mediation analysis requires a number of assumptions for which sensitivity analyses have been developed. First, causal mediation analysis allows for an exposure-mediator interaction, so as a sensitivity analysis, we included an exposure-mediator interaction term in the outcome models to examine whether the natural effect estimates changed [54]. The results of this analysis informed the decision of whether to include exposure-mediator interaction term in all regression models in the main analysis. Second, mediation analysis depends on the assumption of no unmeasured confounding between the exposure and mediator, exposure and outcome, and mediator and outcome (sequential ignorability). Imai et al. have proposed sensitivity analyses to examine this assumption [50,51] but they are not currently supported for count outcome models (i.e., Poisson regression). The main analysis was repeated with DMFS as an outcome. Because we have two proposed mediators in this study, we assumed they were independent for simplicity of the analysis. However, this assumption of independence may not be valid given that convenience stores shopping and SSB intake are likely associated. Violations in this assumption would result in unaddressed confounding in the mediation estimates, which cannot be corrected for using basic mediation methods due to collider stratification bias [55]. Based on descriptive results that suggested that SSB intake may not have a linear relationship with tooth decay, we ran an additional analysis examining the SSB intake mediator as a binary variable.

## Results

### Sample characteristics

Of the 452 study participants, 27 were excluded for missing data on the exposure variable (food insecurity). The final sample size was 425. The mean age of children in the study was 10 years (SD = 3), 53% were female, 58% were Black or African-American, 33% were Hispanic, 84% were publicly insured, and 13% did not have any dental care in the past 12 months (**Table 1**). About 18% of caregivers did not complete high school, 24% completed high school/ received a GED, 32% completed some college or vocational training, and 25% completed college or a higher degree. Annual household income was less than $24,000 for 40% of the sample, $24,000 to $48,000 for 38% of the sample, and more than $48,000 for 25% of the sample. Nearly one-half (44%) of caregivers used food assistance.

About one-half (55%) of the children lived in a food-insecure household. The mean number of decayed tooth surfaces was 0.87 (SD = 1.99; median = 0; IQR = 0,1) and 29% had at least one decayed tooth surface. The mean intake of SSB intake was 17 fl oz/day (SD = 35; median = 7; IQR = 3, 17. About 10% reported 0 fl oz of SSB intake per day, 42% had >0 to 8 fl oz/day, 21% had >8 to 16 fl oz/day, and 26% had >16 fl oz/day. Almost one-half of the sample (45%) reported that they shopped at convenience store 0 to <1 times/month and 37% reported that they shopped at a convenience store 1 time/month to 1 time/week. About 18% of participants reported that they frequently shopped at a convenience store (≥2 times/week).

Children in food-insecure households tended to have higher SSB intake and had a higher prevalence of shopping at a convenience ≥2 times/week than children in food-secure households (23 fl oz/day vs. 11 fl oz/day and 24% vs. 11%, respectively). In addition, living in a food-insecure household was more common among children whose caregivers identified them as Black or African-American, among households with lower annual incomes, and among households who used food assistance.

### Association of food insecurity, SSB intake, and frequent convenience store shopping with tooth decay

After adjusting for confounders, children in food-insecure households had 1.60 times more decayed tooth surfaces than children in food-secure households (95% CI 0.89, 2.88; p = .12) (**Table 2**). Log-transformed SSB intake was also associated with decayed tooth surfaces (mean ratio 1.16; 95% 0.93, 1.46; p = .19). Compared to children who reported 0 fl/oz of SSB intake per day, those who reported >16 fl oz/day had 1.70 more decayed tooth surfaces (95% CI 0.70, 4.15; p = .24). Children in households that frequently shopped at a convenience store (≥2 times/week) had 2.75 times more decayed tooth surfaces than children in households that shopped <2 times/week at a convenience stores (95% CI 1.61, 4.67; p = < .001).

### Mediation analyses

The natural indirect effect of the impact of household food insecurity on tooth decay mediated through log-transformed SSB intake was 0.05 decayed tooth surfaces (95% -0.06, 0.22; p = .35), which indicates that about 9.9% of the influence of food insecurity on tooth decay occurred through SSB intake (**Table 3**). The natural indirect effect of the impact of household food insecurity on tooth decay mediated through frequent convenience store shopping was 0.64 decayed tooth surfaces (95% CI -0.29, 1.68; p = .33), which indicates that about 22.4% of the food insecurity–tooth decay relationship is mediated through frequent convenience store shopping.

**Table 1. Characteristics of children, ages 5 to 16 years, in Seattle and South King County, Washington stratified by household food insecurity in 2018 (N = 425).**

| Characteristic | Food-secure household N = 189 Mean (SD) or n (%) | Food-insecure household N = 236 Mean (SD) or n (%) | Overall N = 425 Mean (SD) or n (%) |
|---|---|---|---|
| **Decayed tooth surfaces** | 0.71 (2.01) | 0.99 (2.01) | 0.87 (1.99) |
| Median (IQR) | 0 (0,0) | 0 (0,1) | 0 (0,1) |
| **>0 decayed tooth surfaces** | 39 (21%) | 82 (35%) | 131 (29%) |
| **SSB intake (fl oz/day) [1]** | 11 (22) | 23 (42) | 17 (35) |
| Median (IQR) | 7 (1,9) | 10 (4,23) | 7 (3,17) |
| Missing | 5 | 3 | 8 |
| **SSB intake category** | | | |
| 0 fl oz/day | 29 (16%) | 13 (5.6%) | 42 (10%) |
| >0 fl oz/day to <8 fl oz/day | 97 (53%) | 80 (34%) | 177 (42%) |
| 8 fl oz/day to <16 flo oz/day | 32 (17%) | 57 (24%) | 89 (21%) |
| >16 fl oz/day | 26 (14%) | 83 (36%) | 109 (26%) |
| Missing | 5 | 3 | 8 |
| **Frequency of convenience store shopping** | | | |
| Never or <1 time/month | 105 (56%) | 82 (36%) | 187 (45%) |
| 1 time/ month to 1 time/week | 61 (33%) | 92 (40%) | 153 (37%) |
| ≥2 times/week | 21 (11%) | 54 (24%) | 75 (18%) |
| Missing | 2 | 8 | 10 |
| **Child age (years)** | 10 (3) | 10 (2) | 10 (3) |
| 5–8 years | 51 (28%) | 68 (29%) | 119 (29%) |
| 9–12 years | 91 (49%) | 113 (49%) | 204 (49%) |
| 13–16 years | 43 (23%) | 50 (22%) | 93 (22% |
| Missing | 4 | 5 | 9 |
| **Child sex** | | | |
| Female | 104 (55%) | 122 (52%) | 226 (53%) |
| Male | 85 (45%) | 113 (48%) | 198 (47%) |
| Missing | 0 | 1 | 1 |
| **Child race[3]** | | | |
| Asian | 5 (2.5%) | 4 (2.6%) | 9 (2.5%) |
| Black | 125 (61%) | 84 (54%) | 209 (58%) |
| Native Hawaiian or Other Pacific Islander | 3 (1.5%) | 4 (2.6%) | 7 (1.9%) |
| White | 30 (15%) | 45 (29%) | 75 (21%) |
| Multi-racial | 41 (20%) | 19 (12%) | 60 (17%) |
| Missing | 32 | 33 | 65 |
| **Child Hispanic ethnicity** | 63 (33%) | 78 (33%) | 141 (33%) |
| **Child insurance type[4]** | | | |
| Public insurance | 137 (77%) | 202 (90%) | 339 (84%) |
| Private insurance | 42 (23%) | 22 (9.8%) | 64 (16%) |
| Missing | 10 | 12 | 22 |
| **Child did not have dental care in past 12 months** | 18 (9.5%) | 38 (16%) | 56 (13%) |
| **Caregiver education** | | | |
| Did not complete high school | 27 (15%) | 50 (21%) | 77 (18%) |
| Completed high school/GED | 48 (26%) | 53 (23%) | 101 (24%) |
| Some college or vocational training | 53 (28%) | 83 (36%) | 136 (32%) |
| Completed college or above | 58 (31%) | 47 (20%) | 105 (25%) |
| Missing | 3 | 3 | 6 |
| **Annual household income** | | | |

*(Continued)*

**Table 1.** (Continued)

| Characteristic | Food-secure household<br>N = 189<br>Mean (SD) or n (%) | Food-insecure household<br>N = 236<br>Mean (SD) or n (%) | Overall<br>N = 425<br>Mean (SD) or n (%) |
|---|---|---|---|
| Less than $24,000 | 55 (32%) | 108 (47%) | 163 (40%) |
| $24,000 to $48,000 | 55 (32%) | 97 (42%) | 152 (38%) |
| More than $48,000 | 64 (37%) | 24 (10%) | 88 (22%) |
| Missing | 15 | 7 | 22 |
| **Uses food assistance[4]** | 61 (32%) | 126 (54%) | 187 (44%) |
| Missing | 1 | 2 | 3 |

IQR, interquartile range; SSB, sugar-sweetened beverage; fl oz, fluid ounces; GED, General Educational Development.

[1] SSB intake was measured via the 20-item Beverage Intake Questionnaire (BEV-Q).

[2] Of the children with reported missing race, 61 (94%) reported Hispanic ethnicity.

[3] All children in the study had insurance with the exception of one, who was set to missing insurance type.

[4] Food assistance used was defined as participation in either Supplemental Nutrition Assistance Program (SNAP) or Special Supplemental Nutrition Program for Women, Infants, and Children (WIC).

**Table 2. Number of decayed tooth surfaces on household food insecurity, SSB intake, and convenience store shopping among children, ages 5 to 16 years, in Seattle and South King County, Washington in 2018.**

| | Decayed tooth surfaces mean (SD) | Mean ratio[1] (95% CI) | p |
|---|---|---|---|
| **Household food insecurity** | | | |
| Food-insecure | 0.99 (2.01) | 1.60 (0.89, 2.88) | .12 |
| Food-secure | 0.71 (2.01) | Ref | |
| **Log-transformed SSB intake (fl oz/day)[2]** | - | 1.16 (0.93, 1.46) | .19 |
| **SSB intake category** | | | |
| 0 fl oz/day | 0.59 (1.40) | Ref | |
| >0 fl oz/day to 8 fl oz/day | 0.88 (2.03) | 1.29 (0.56, 2.97) | .55 |
| >8 fl oz/day to 16 flo oz/day | 0.63 (1.29) | 1.09 (0.48, 2.50) | .84 |
| >16 fl oz/day | 1.18 (2.62) | 1.70 (0.70, 4.15) | .24 |
| **Any SSB intake (>0 fl oz/day)** | | | |
| No | 0.59 (1.40) | Ref | |
| Yes | 0.91 (2.09) | 1.32 (0.62, 2.82) | .47 |
| **Frequency of convenience store shopping** | | | |
| Never or <1 time/month | 0.76 (1.91) | Ref | |
| 1 time/month to 1 time/week | 0.74 (1.73) | 0.74 (0.42, 1.32) | .31 |
| ≥2 times/week | 1.48 (2.66) | 2.39 (1.35, 4.22) | .003 |
| **Frequent convenience store shopping** | | | |
| <2 times/week | 0.75 (1.82) | Ref | |
| ≥2 times/week | 1.48 (2.66) | 2.75 (1.61, 4.67) | < .001 |

SSB, sugar-sweetened beverage; SD, standard deviation; CI, confidence interval, fl oz, fluid ounces.

[1] Six Poisson regression with robust standard errors were used to estimate the mean ratios, 95% CI, and p-values. The mean ratio can be interpreted as the ratio of the average of number of decayed tooth surfaces between two groups. Models were adjusted for child age, child race, child Hispanic ethnicity, child insurance type, caregiver education, annual household income, food assistance use, and number of tooth surfaces was included as a covariate in the regression models.

[2] SSB intake was measured via a 20-item beverage questionnaire. For this analysis, it was transformed as natural log(x +1) to account for right skew and to include participants who reported 0 fl oz of SSB intake.

**Table 3. Mediating effects of SSB intake and frequent convenience store shopping in the household food insecurity–tooth decay relationship for children, ages 5 to 16 years, in Seattle and South King County, Washington in 2018.**

| Proposed Mediator | Decayed tooth surfaces | | | | | | |
|---|---|---|---|---|---|---|---|
| | Total Effect[1] (95% CI) | p | Natural indirect effect[2] (95% CI) | p | Natural direct effect[3] (95% CI) | p | % Mediated |
| **Log-transformed SSB intake, (fl oz/day)[4]** | 0.45 (-0.18, 1.05) | .14 | 0.05 (-0.06, 0.22) | .35 | 0.39 (-0.24, 0.97) | .18 | 9.9% |
| **Any SSB intake (>0 fl oz/day)** | 0.53 (-0.21, 1.40) | .14 | 0.03 (-0.16, 0.36) | .88 | 0.50 (-0.22, 1.23) | .13 | 0.7% |
| **Frequent convenience store shopping[5]** | 1.16 (-0.13, 2.68) | .09 | 0.64 (-0.29, 1.68) | .33 | 0.52 (-0.26, 1.40) | .16 | 22.4% |

SSB, sugar-sweetened beverage; CI, confidence interval, fl oz, fluid ounces.

[1] The total effect can be interpreted as the differences in the average number of decayed tooth surfaces between children in food-secure and food-insecure households.

[2] The natural indirect effect can be interpreted as the impact of food insecurity on the number of decayed tooth surfaces that operates through the mediator.

[3] The natural direct effect can be interpreted as the impact of food insecurity on the number of decayed tooth that does not operate through the mediator.

[4] SSB intake was measured via a 20-item beverage questionnaire. For this analysis, it was transformed as natural $\log(x+1)$ to account for right skew and to include participants who reported 0 fl oz of SSB intake.

[5] Frequent convenience store shopping was defined as $\geq 2$ times/week.

Poisson regression was used to model decayed tooth surfaces and frequent convenience store shopping as outcomes and linear regression was used to model SSB intake as an outcome. Estimates were adjusted for child age, child race, child Hispanic ethnicity, child insurance, caregiver education, annual household income, food assistance use, and number of tooth surfaces was included as a covariate in the regression models (when tooth decay was the outcome). Full model specification is shown in Appendix II. All estimates for the mediation analyses are reported on the additive scale (mean differences in the number of decayed tooth surfaces) at the mean level of confounders, or the most frequent level for categorical confounders. Robust standard errors were used to generate confidence intervals.

## Sensitivity analyses

Including an interaction between exposure and mediators in the mediation analyses did not meaningfully influence the mediation estimates, so an interaction term was omitted from all main analyses (S1 Table). Food insecurity, SSB intake, and frequent convenience stores shopping were not associated with DMFS (S2 and S3 Tables).

## Discussion

In this study, we examined whether the household food insecurity–tooth decay relationship was mediated by SSB intake and frequent convenience store shopping for children in lower-income households in King County, Washington state. We found that household food insecurity, SSB intake, and frequent convenience store shopping were associated with higher levels of mean decayed tooth surfaces. Specifically, children in lower-income households that reported shopping at a convenience store ≥2 times/week had 2.4 times more decayed tooth surfaces than children in low-income households that shopped at a convenience store less frequently. According to mediation analyses, SSB intake mediated 10% of the impact of food insecurity on tooth decay and frequent convenience store shopping mediated about 22%, although estimates were not statistically significant. Children with low SES are at a high risk for tooth decay and evidence-based public health interventions are needed to address disparities. Potential interventions include food insecurity prevention, SSB taxes, and community education programs that improve food and beverage purchasing decisions in tandem with structural changes that improve access to healthy and affordable food and beverage options in low-resource communities.

Consistent with the literature that has reported that household food insecurity is a risk factor for childhood tooth decay [10–15], we found that household food-insecurity was associated with 1.6 times more decayed tooth surfaces among the low-income children in this study. We also found that higher levels of SSB intake tended to be associated with higher levels of tooth decay, which aligns with the biological understanding of how sugar acts as a substrate for

bacteria that cause tooth decay [17]. One new finding in this study was that frequent convenience store shopping was strongly associated with tooth decay. Past research has suggested that convenience store shopping may have a modest but positive association with other adverse chronic health outcomes in children, such as obesity [56–59]. Convenience stores may be a particularly relevant setting to target for interventions aimed at addressing childhood oral health disparities because it is a high-sugar environment and a common source of not only SSBs [24,25], but also of nutrient-poor foods high in added sugars [60]. Targeting behaviors alone is not adequate as structural factors also shape the distribution of convenience stores and supermarkets and the price of healthy foods in communities. Policies should aim to improve access to a variety of healthy and affordable foods options in all stores in low-income communities.

Although estimated mediation effects were not statistically significant in this study, findings suggested that SSB intake and frequent convenience store shopping transmitted about 10% and 22% of the impact of food insecurity onto tooth decay, respectively. The one known study that examined mediators in the household food insecurity–tooth decay relationship tested diet quality but did not complete mediation analyses because the association between diet quality and tooth decay was not statistically significant [14]. According to previous research, food-insecure households have higher SSB intake [17,19–23] and shop at convenience stores more frequently [28–30], which provides support that each may be a relevant mediator in the food insecurity–tooth decay relationship. However, the lack of statistically significant mediation estimates in this study could mean that SSB intake and frequent convenience store shopping are not truly mediators and the effect of food insecurity on tooth decay occurs via other mechanisms. For example, the impact of food security on tooth decay could be attributed to socioeconomic pathways, given that food security is a marker of SES. It is also likely that the lack of statistically significant mediation effects in this study could be explained by the relatively small study sample size that had insufficient power to detect precise mediation estimates given that confidence intervals were wide but mostly covered effect sizes that would indicate a true relationship in the hypothesized direction. Given the lack of certainty in the study findings, follow-up studies using sufficient sample sizes and a longitudinal study design are needed to verify whether SSB intake and frequent convenience store shopping are truly mediators in the food insecurity–tooth decay relationship.

Regardless of whether SSB intake and frequent convenience store shopping are relevant mediators of the food insecurity–tooth decay relationship, they were each associated with tooth decay which indicates there are opportunities to address oral health disparities through policy and program interventions that impact SSB intake and access to healthy foods at convenience stores. One policy intervention is SSB taxes which not only aim to reduce SSB intake but also divert tax revenue to fund programs for nutrition improvement in low-income populations [61]. According to a meta-analysis of SSB taxes across 62 studies in cities and national governments, SSB taxes were associated with reductions in SSB sales [62]. In Seattle, Washington, which is a region captured in the current study, a SSB tax in January 2018 led to a 22% decline in the volume of SSBs purchased [63]. However, there was no corresponding decrease in SSB intake directly associated with the tax [64]. Regardless, in the two-year period after the tax was enacted, support for the tax increased among low-income residents [65] and the perceived negative health impacts of SSBs increased among low-income residents [66], which suggests that SSB taxes could have positive spillover effects on the incidence of tooth decay among Seattle children. While no experimental studies have examined whether SSB taxes improve oral health, numerous simulation and model-based studies have predicted the potential public health benefits of a SSB tax on oral health outcomes [67]. Predictions in high-income countries like the U.S. show that SSB taxes could lead to reductions in the incidence of tooth decay and

reduced spending on treatment costs [67]. An experimental study could identify whether SSB taxes have tangible impacts on tooth decay and can inform evidence-based policy decisions.

There is also opportunity to intervene on oral health outcomes through the food environment [68,69]. While the association of neighborhood access to convenience stores and child SSB intake is equivocal [70–72] and many families travel outside of their neighborhoods to obtain food [73], low-resource neighborhoods still offer the fewest food-purchasing establishments and convenience stores are common [26,74]. In addition, food-insecure households are more likely than food-secure households to not own their own vehicle to use for shopping (they rely on others for transportation or public transportation) and travel shorter distances when shopping [28–30], which means that the neighborhood food environment may be particularly relevant for food-insecure families. At least one study in Pittsburgh, Pennsylvania, has shown added sugar intake dropped after a supermarket was opened in a community with limited access to healthy foods [75]. Ensuring the availability of a variety of food-purchasing establishments that stock healthy and affordable food and beverage options is a step to support healthy food purchasing in low-resource communities. However, additional efforts are also necessary. These efforts include enforcement of living wages, regulations on food advertising, and population-based education programs that publicize the potential negative effects of excess added sugar intake [76], provide education about healthier food and beverage choices, meal preparation resources, and information about how to introduce new foods to picky children.

While not statistically significant, our study findings suggested that about 11% and 20% of the impact of food insecurity on tooth decay was transmitted through SSB intake and frequent convenience store shopping, respectively. Therefore, interventions that reduce SSB intake and promote intake of healthy foods could be a particularly effective way to address oral health disparities in food-insecure communities. For instance, SNAP participants can purchase SSBs with benefits and as of 2016, SSBs are among the ten most commonly-purchased items among SNAP participants [77]. Simulation research suggests that SSB-purchasing restrictions in SNAP could lead to an 8% decrease in the number of decayed teeth among children in SNAP-participating households [78]. Furthermore, policies that implement incentives for healthy food in addition to SSB restrictions may prove to be the most in fruitful in making public health improvements [79]. However, SSB-purchasing restrictions in SNAP remain controversial and there is strong corporate lobbying against such policy changes [80]. Improving food purchasing habits in food-insecure households could also improve oral health disparities. For example, among food-insecure households participating in SNAP, point-of-sale incentives have been linked to higher vegetable and fruit expenditures [81–83].

## Strengths and limitations

There are strengths to this study. First, the study focused on a lower-income racially-diverse sample of children, a population that experiences some of the greatest oral health disparities and stands to gain the most benefit from public health interventions. Second, we incorporated recently-developed causal mediation techniques. These approaches improve on traditional mediation techniques by allowing for an exposure-mediator interaction, analysis of a broader variety of exposures, mediators and outcomes, and moves away from a reliance on statistical significance which is required in the traditional Baron Kenny approach [84]. We also add to the growing literature that has identified food insecurity as a relevant oral health risk factor and attempt to understand underlying mechanisms which is key to developing successful evidence-based public health interventions.

Limitations should also be considered when interpreting the findings of this study. First, because our data are cross-sectional we cannot directly infer the temporality of our study

measures. All published research reporting the food insecurity–tooth decay relationship has relied on cross-sectional data which emphasizes the need for a well-designed longitudinal study that can more definitively discern temporality. Second, the relatively small study sample size likely contributed to insufficient power for the mediation analyses and may be a primary explanation for the lack of statistically significant mediation findings. According to simulations conducted by Fritz and Mackinnon, at least 500 observations are needed to detect significant direct and indirect effects when studying small effects [85], such as those that would be expected when studying decayed tooth surfaces as an outcome. Future longitudinal and intervention studies should ensure that an adequate sample size is met which can be achieved using newly-developed power calculations for mediation analyses [86,87]. Third, it is possible there was measurement error in SSB intake due to social desirability bias, especially for adolescents who self-reported their intake and the measure of convenience store given that adults reporting shopping behaviors for the household may have not captured adolescent's independent shopping behaviors. While the BEV-Q used in this study was found to be valid and reliable among adolescents [40–42], this potential measurement error should be considered when interpreting the study findings. Lastly, we were unable to conduct the sensitivity analyses that have been developed for causal mediation analyses because they are not currently supported for models with count outcomes. While we cannot examine how robust the present findings are to the assumptions of causal mediation analysis, future work should aim to incorporate sensitivity analyses as they are developed.

## Conclusion

This study examined the extent to which SSB intake and frequent convenience store shopping mediated the household food insecurity–tooth decay relationship among a racially-diverse lower-income sample of children in King County, Washington state. Consistent with previous literature, household food insecurity and SSB intake were associated with higher levels of tooth decay. We also found that frequent convenience store shopping was associated with 2.4 times more decayed tooth surfaces. Although mediation estimates were not statistically significant which may have resulted from the relatively small study sample size, SSB intake and frequent convenience store mediated about 10% and 22% of the impact of household food insecurity on tooth decay. Concerted policy and public health efforts are needed to address tooth decay disparities among children with low SES, including enforcement of living wages, improvements in the access and affordability of healthy foods in low-income neighborhoods, SSB taxes, and community education efforts that improve food and beverage purchasing decisions in low-resource communities.

## Supporting information

**S1 Checklist. Table 1.** STROBE-nut: An extension of the STROBE statement for nutritional epidemiology.
(DOCX)

**S1 File. Confounder selection.**
(DOCX)

**S2 File. Mediator models.**
(DOCX)

**S1 Table. Mediating effects of SSB intake and frequent convenience store shopping in the household food insecurity–tooth decay relationship allowing for an interaction between food insecurity and mediator, for children, aged 5 to 16 years, in Seattle and South King**

**County, 2018.**
(DOCX)

**S2 Table. Number of DMFS on household food insecurity, SSB intake, and frequent convenience store shopping among children, aged 5 to 16 years, in Seattle and South King County, 2018.**
(DOCX)

**S3 Table. Mediating effects of SSB intake and frequent convenience store shopping in the household food insecurity–DMFS relationship for children, aged 5 to 16 years, in Seattle and South King County, 2018.**
(DOCX)

**S1 Data. SeaSAW_OH_data_restricted.**
(XLSX)

**S2 Data. SeaSAW_OH_data_restricted_codebook.**
(XLSX)

## Author Contributions

**Conceptualization:** Courtney M. Hill, Donald L. Chi, Jessica C. Jones-Smith, Nadine Chan, Brian E. Saelens, Christy M. McKinney.

**Formal analysis:** Courtney M. Hill.

**Funding acquisition:** Brian E. Saelens, Christy M. McKinney.

**Investigation:** Donald L. Chi.

**Methodology:** Courtney M. Hill, Lloyd A. Mancl.

**Writing – original draft:** Courtney M. Hill.

**Writing – review & editing:** Donald L. Chi, Lloyd A. Mancl, Jessica C. Jones-Smith, Nadine Chan, Brian E. Saelens, Christy M. McKinney.

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
