## [Decision Letter · Decision Letter 0]

22 May 2023

PONE-D-23-10186Sugar-sweetened beverage intake and convenience store shopping as mediators of the food insecurity–tooth decay relationship among low-income children in Washington StatePLOS ONE

Dear Dr. Hill,

Thank you for submitting your manuscript to PLOS ONE. After careful consideration, we feel that it has merit but does not fully meet PLOS ONE’s publication criteria as it currently stands. Therefore, we invite you to submit a revised version of the manuscript that addresses the points raised during the review process.

While I encourage you to respond to each comment made by the four peer reviewers, you are not required to follow a suggestion that asks you to cite specific publications in the manuscript. Please focus on other comments when preparing for a revision.

We look forward to receiving your revised manuscript.

Kind regards,

Boyen Huang, DDS, MHA, PhD

Academic Editor

PLOS ONE

Journal Requirements:

Reviewers' comments:

Reviewer's Responses to Questions

**Comments to the Author**

1. Is the manuscript technically sound, and do the data support the conclusions?

Reviewer #1: Yes

Reviewer #2: Yes

Reviewer #3: Yes

Reviewer #4: Yes

2. Has the statistical analysis been performed appropriately and rigorously? 

Reviewer #1: Yes

Reviewer #2: Yes

Reviewer #3: Yes

Reviewer #4: Yes

3. Have the authors made all data underlying the findings in their manuscript fully available?

Reviewer #1: Yes

Reviewer #2: Yes

Reviewer #3: Yes

Reviewer #4: No

4. Is the manuscript presented in an intelligible fashion and written in standard English?

Reviewer #1: Yes

Reviewer #2: Yes

Reviewer #3: Yes

Reviewer #4: Yes

5. Review Comments to the Author

Reviewer #1: Dear Authors, this paper about sugar-sweetened beverage intake and convenience store shopping as mediators of the food insecurity–tooth decay relationship among low-income children in Washington State is really interesting and well performed. I am sure it will help both clinicians and researchers to improve their job. Overall the paper is well written, nevertheless some issues need to be solved before its final publication in the journal.

Abstract: please divide abstract into introduction, materials and methods, results, conclusion.

Introduction: this part of an article is really important and it helps the reader to deep into the subject.

Materials and methods and results are overall well described an easy to understand.

Discussion: this part is well written but i would suggest to enlarge a little the discussion focusing also in a general population.

Reviewer #2: This research topic is a creative approach to the relationship between food insecurity and dental caries which is attracting global attention and problems.

It is difficult to determine the relevance of dental caries in that they are cumulative diseases and do not progress in a short period of time.

In this respect, it is interesting to study using causal mediation techniques.

The limitations and questions about this study are well described in the manuscript.

However, sugar-containing drinks have no mechanism to cause tooth decay so it is recommended to add this part.

It is also recommended that you write additional criteria for the sample size in the methods.

Thank you.

Reviewer #3: Thank-you for the opportunity to review this interesting paper. It is a well written manuscript with an interesting and valuable analysis that attempts to better understand the interaction of food insecurity, SSB consumption and convenience store shopping on dental caries. I have made a few comments and suggestions below>

Introduction

It would be useful to state how food insecurity is defined.

It may be beneficial to expand on the rationale for using mediation analysis, particularly as it may be unfamiliar to some readers.

Methods

The authors mention that the data was collected at a period that would not overlap with the introduction of the SSB tax. For completeness it would be useful to include the date that the SSB tax was introduced.

One of the eligibility criteria was ‘annual household income <312% of the Federal Poverty Level.’ This is a weirdly specific percentage, but presumably has some rationale?

One of the eligibility criteria was ‘child consumed SSBs’. Yet 42 of the children in the final sample were categorised as having a SSB intake of 0 fl oz/day. Shouldn’t these children have been excluded from the study?

It might be useful to make a comment on the validity and reliability of the BEV-Q for assessing beverage consumption (and particularly the difference if any between caregiver assessment for children <11 years and self-assessment for children >11 years).

For the measure ‘frequency of convenience store shopping’, was this assessed as a family unit and reported by the caregiver for all participants, or was it assessed by the caregiver for children <11 years and by self-assessment for children >11 years? I’m wondering whether children >11 years might independently purchase SSBs (or other foods/drinks) from convenience stores independently of the family.

Reviewer #4: This study of sugar-sweetened beverage intake and convenience store shopping as mediators of

the food insecurity–tooth decay contributes to a small but growing body of evidence related to structural factors linked to tooth decay. The paper is well organized, and the analysis clearly described. However, in examining Table 1 - there are some discrepancies. The child race statistics appear to be in the wrong columns. There is also inconsistency in how variables are reported which diminishes clarity. Some variables include the entire N, while others do not. Reporting age by age groups would provide more clarity on the age characteristics of the sample rather than just reporting the mean. Finally, some limitations of having 11 to 16 year old's self-report their beverage intake, e.g., social desirability bias, should be discussed.

6. PLOS authors have the option to publish the peer review history of their article (what does this mean?). If published, this will include your full peer review and any attached files.

Reviewer #1: No

Reviewer #2: No

Reviewer #3: No

Reviewer #4: No

---

## [Author Response · Author response to Decision Letter 0]

6 Jul 2023

Academic Editor- Journal Requirements:

Style requirements have been checked and we ensured that our manuscript meets the requirements. 

The information in the Financial Disclosure section should be as follows: “This work was support by funds from Seattle Children’s Research Institute and ARCORA – The Foundation of Delta Dental of Washington. Funding for the Sugary Beverage Tax evaluation in Seattle was provided through ordinance by the City of Seattle. Elected representatives and city staff did not influence the evaluation findings or interpretation of findings.”

We do not wish to make changes to the Data Availability Statement.

Captions for the supporting information files were included.

The reference list has been reviewed and no changes were made.

Reviewer #1: Dear Authors, this paper about sugar-sweetened beverage intake and convenience store shopping as mediators of the food insecurity–tooth decay relationship among low-income children in Washington State is really interesting and well performed. I am sure it will help both clinicians and researchers to improve their job. Overall the paper is well written, nevertheless some issues need to be solved before its final publication in the journal.

Thank you for reviewing our paper and providing comments.

Abstract: please divide abstract into introduction, materials and methods, results, conclusion.

We have added subheadings to the abstract per your comment.

Introduction: this part of an article is really important and it helps the reader to deep into the subject. Materials and methods and results are overall well described an easy to understand.

Discussion: this part is well written but i would suggest to enlarge a little the discussion focusing also in a general population.

Thank you for your comment. We compared our results for the association of household food insecurity and tooth decay in the discussion to other studies in the general population and state that findings are consistent. Specifically, we cite three studies that used the National Health and Nutrition Examination Survey (NHANES), which is a representative cross-section of the general U.S. population of children. We also discuss the implications of a sugar-sweetened beverage tax more generally and cite a meta-analysis that discusses evaluations of a such a tax across multiple national and municipal governments to expand the interpretation of this study. 

Reviewer #2: This research topic is a creative approach to the relationship between food insecurity and dental caries which is attracting global attention and problems.

Thank you for reviewing our paper and providing comments. 

It is difficult to determine the relevance of dental caries in that they are cumulative diseases and do not progress in a short period of time.

In this respect, it is interesting to study using causal mediation techniques.

The limitations and questions about this study are well described in the manuscript.

However, sugar-containing drinks have no mechanism to cause tooth decay so it is recommended to add this part.

We have revised the manuscript to try and make this clearer. In the introduction, we state that, “excess intake of added sugars, including those from SSBs is causally associated with the development of tooth decay.[17,18]” (lines 51-53). In the discussion, we further state that, “We also found that higher levels of SSB intake tended to be associated with higher levels of tooth decay, which aligns with the biological understanding of how sugar acts as a substrate for bacteria that cause tooth decay.[17]” (lines 295-296) To make it clear that the potential mechanism by which sugar-sweetened beverages (SSBs) can influence tooth decay is through sugar acting as a substrate for oral bacteria that cause tooth decay, we expanded the statement in the introduction: “Excess intake of added sugars, including those from SSBs, is causally associated with the development of tooth decay, because sugar acts as a substrate for oral bacteria that cause tooth decay.[17,18]” (lines 51-53)

It is also recommended that you write additional criteria for the sample size in the methods.

We have stated the study criteria in the methods: “There were five eligibility criteria for participation in the additional study: (1) the child was in 1st to 9th grade (5 to 16 years old); (2) the child consumed SSBs based on parent report; (3) the annual household income was <312% of the Federal Poverty Level (FPL); (4) the caregiver was ≥18 years old; and (5) the caregiver spoke English, Spanish, Somali or Vietnamese. The screening item about SSBs was, “Does your child ever drink sugary beverages like: regular soda/pop (such as Coke or Sprite), fruit-flavored drinks (like Sunny Delight), coffee or tea drinks with added sugar (like Starbucks Frappucinnos, Arizona Iced Tea, Chai Tea, bubble tea), or regular sports drinks or energy drinks (such as Gatorade or Red Bull)?”. The SeaSAW study had slightly different child age criterion; namely, children were required to be either 7-10 or 12-17 years old, but otherwise had the same inclusion criteria.[33]”. (lines 93-103) We also state in the study results how the final sample size was arrived at: “Of the 452 study participants, 27 were excluded for missing data on the exposure variable (food insecurity). The final sample size was 425.” (lines 229-230)

Thank you.

Reviewer #3: Thank-you for the opportunity to review this interesting paper. It is a well written manuscript with an interesting and valuable analysis that attempts to better understand the interaction of food insecurity, SSB consumption and convenience store shopping on dental caries. I have made a few comments and suggestions below>

Thank you for reviewing our paper and providing comments.

Introduction

It would be useful to state how food insecurity is defined.

We agree, and have added the United States Department of Agriculture’s definition of food insecurity to the introduction: “According to the United States Department of Agriculture, food insecurity is a household-level economic and social condition of limited or uncertain access to adequate food [16]” (lines 38-40)

It may be beneficial to expand on the rationale for using mediation analysis, particularly as it may be unfamiliar to some readers.

We have included additional rationale for using mediation analysis in the last paragraph of the introduction: “Mediation analysis was used in this analysis because it is a tool used to discover and test possible causal relationships by teasing apart direct effects of an exposure on an outcome from indirect effects that occur through a third mediator variable.” (lines 71-74)

Methods

The authors mention that the data was collected at a period that would not overlap with the introduction of the SSB tax. For completeness it would be useful to include the date that the SSB tax was introduced.

We revised the methods to state when the tax went into effect (January 2018) (lines 84-85) Measuring baseline dental caries within 4 to 6 months of tax implementation reasonably estimates caries status pre-tax given the time progression of caries. Change in caries status typically requires at least 2 years of follow-up.

One of the eligibility criteria was ‘annual household income <312% of the Federal Poverty Level.’ This is a weirdly specific percentage, but presumably has some rationale?

There are various ways to define lower income. <312% of the Federal Poverty Level corresponds to cut-off points used by Washington state policies for public health insurance. For example, Apple Health for Kids (Washington state Medicaid program/ the public-insurance program in Washington state) uses 312% of the Federal Poverty level as the upper limit for eligibility for plans. 

One of the eligibility criteria was ‘child consumed SSBs’. Yet 42 of the children in the final sample were categorised as having a SSB intake of 0 fl oz/day. Shouldn’t these children have been excluded from the study?

Children were screened into the study based on a single item that required an affirmative answer: “The screening item about SSBs was, “Does your child ever drink sugary beverages like: regular soda/pop (such as Coke or Sprite), fruit-flavored drinks (like Sunny Delight), coffee or tea drinks with added sugar (like Starbucks Frappucinnos, Arizona Iced Tea, Chai Tea, bubble tea), or regular sports drinks or energy drinks (such as Gatorade or Red Bull)?”.” (lines 97-101) Forty-two children in the final sample had a reported intake of SSB of 0 fl/oz per day based on responses to BEV-Q which captures typical intake in the past 30 days. Therefore, it is possible that while these children do sometimes drink SSB, typical intake in the past 30 days was about 0 fl/oz per day. For this reason, these children were not excluded from the final sample (see lines 100-102).

It might be useful to make a comment on the validity and reliability of the BEV-Q for assessing beverage consumption (and particularly the difference if any between caregiver assessment for children <11 years and self-assessment for children >11 years).

We included additional details about the validity and reliability testing of the BEV-Q in the methods: “The original BEV-Q was tested in adolescents and shown to have high validity against multiple 24-hour dietary recalls, high reliability across repeat measurements, and readability scores appropriate for individuals with a fourth grade education or higher.[40–42]” (lines 123-126) This was the justification for having parents report on younger children’s intake and those above 11 years to self-report.

For the measure ‘frequency of convenience store shopping’, was this assessed as a family unit and reported by the caregiver for all participants, or was it assessed by the caregiver for children <11 years and by self-assessment for children >11 years? I’m wondering whether children >11 years might independently purchase SSBs (or other foods/drinks) from convenience stores independently of the family.

This was reported by the caregiver for family-level grocery shopping patterns regardless of child age. It is certainly possible that children >11 years independently purchase SSBs and other drinks from the convenience store and this is a limitation of the measure. We added this as a limitation in addition to potential social desirability bias that adolescents may have exhibited when reporting their SSB intake on the BEV-Q: “Third, it is possible there was measurement error in SSB intake due to social desirability bias, especially for adolescents who self-reported their intake and in the measure of convenience store given that adults reporting shopping behaviors for the household may have not captured adolescent’s independent shopping behaviors. While the BEV-Q used in this study was found to be valid and reliable among adolescents,[40-42] this potential measurement error should be taken into account when interpreting the study findings.” (lines 400-406)

Reviewer #4: This study of sugar-sweetened beverage intake and convenience store shopping as mediators of

the food insecurity–tooth decay contributes to a small but growing body of evidence related to structural factors linked to tooth decay. The paper is well organized, and the analysis clearly described.

Thank you for reviewing our paper and providing comments.

However, in examining Table 1 - there are some discrepancies. The child race statistics appear to be in the wrong columns. 

Thank you for pointing out the error for race in Table 1 (the “food-secure” and “overall” column had been switched and are now corrected). 

There is also inconsistency in how variables are reported which diminishes clarity. Some variables include the entire N, while others do not. 

All variables in Table 1 are reported for the total number of participants with available non-missing data. Missingness is minor across most variables (<5%) with the child race exhibiting the highest level of missing data (n=65, 15%). Please also note that we added an explanation to the Table 1 notes that of the 65 children with missing race data, 61 (94%) reported Hispanic ethnicity.

Reporting age by age groups would provide more clarity on the age characteristics of the sample rather than just reporting the mean. 

Age groups have been added to Table 1 to improve clarity.

Finally, some limitations of having 11 to 16 year old's self-report their beverage intake, e.g., social desirability bias, should be discussed.

We have added a limitation about our measures of both self-reported SSB intake for 11-16 year olds and the lack of measurement about independent convenience store shopping for the 11-16 year old age groups to the limitations: “Third, it is possible there was measurement error in SSB intake due to social desirability bias, especially for adolescents who self-reported their intake and in the measure of convenience store given that adults reporting shopping behaviors may have not captured adolescent’s independent shopping behaviors. While the BEV-Q used in this study was found to be valid and reliable among adolescents,[40-42] this potential measurement error should be taken into account when interpreting the study findings.” (lines 400-406)

---

## [Decision Letter · Decision Letter 1]

7 Aug 2023

Sugar-sweetened beverage intake and convenience store shopping as mediators of the food insecurity–tooth decay relationship among low-income children in Washington State

PONE-D-23-10186R1

Dear Dr. Hill,

We’re pleased to inform you that your manuscript has been judged scientifically suitable for publication and will be formally accepted for publication once it meets all outstanding technical requirements.

Kind regards,

Boyen Huang, DDS, MHA, PhD

Academic Editor

PLOS ONE

Additional Editor Comments (optional):

Reviewers' comments:

Reviewer's Responses to Questions

**Comments to the Author**

1. If the authors have adequately addressed your comments raised in a previous round of review and you feel that this manuscript is now acceptable for publication, you may indicate that here to bypass the “Comments to the Author” section, enter your conflict of interest statement in the “Confidential to Editor” section, and submit your "Accept" recommendation.

Reviewer #2: All comments have been addressed

Reviewer #3: All comments have been addressed

Reviewer #4: All comments have been addressed

2. Is the manuscript technically sound, and do the data support the conclusions?

Reviewer #2: Yes

Reviewer #3: Yes

Reviewer #4: (No Response)

3. Has the statistical analysis been performed appropriately and rigorously? 

Reviewer #2: Yes

Reviewer #3: Yes

Reviewer #4: (No Response)

4. Have the authors made all data underlying the findings in their manuscript fully available?

Reviewer #2: Yes

Reviewer #3: Yes

Reviewer #4: (No Response)

5. Is the manuscript presented in an intelligible fashion and written in standard English?

Reviewer #2: Yes

Reviewer #3: Yes

Reviewer #4: (No Response)

6. Review Comments to the Author

Reviewer #2: I think this study will appeal to readers.

Previous review content has been appropriately amended.

The manuscript has been improved.

Reviewer #3: Thank-you for the opportunity to review this re-submission. The authors have adequately addressed the concerns of the reviewers. I have no additional comments.

Reviewer #4: (No Response)

7. PLOS authors have the option to publish the peer review history of their article (what does this mean?). If published, this will include your full peer review and any attached files.

Reviewer #2: No

Reviewer #3: No

Reviewer #4: No

---

## [Editor Report · Acceptance letter]

4 Sep 2023

PONE-D-23-10186R1 

Sugar-sweetened beverage intake and convenience store shopping as mediators of the food insecurity–tooth decay relationship among low-income children in Washington State 

Dear Dr. Hill:

I'm pleased to inform you that your manuscript has been deemed suitable for publication in PLOS ONE. Congratulations! Your manuscript is now with our production department. 

Kind regards, 

on behalf of

Dr Boyen Huang 

Academic Editor

PLOS ONE